# Psychosocial working conditions as determinants of slips and lapses, and poor social interactions with patients among medical assistants in Germany: A cohort study

**Viola Mambrey, Adrian Loerbroks***

Institute of Occupational, Social and Environmental Medicine, Centre for Health and Society, Medical Faculty and University Hospital Düsseldorf, Heinrich Heine University Düsseldorf, Düsseldorf, Germany

* Adrian.Loerbroks@uni-duesseldorf.de

## Abstract

### Objective

We sought to examine the relationship of unfavorable psychosocial working conditions with slips and lapses and poor patient interaction as well as potential intermediate factors among medical assistants (MAs) in Germany based on prospective data.

### Methods

We used data from 408 MAs from a 4-year cohort study (follow-up: 2021). At baseline, psychosocial working conditions were assessed by the established effort-reward-imbalance questionnaire and a MA-specific questionnaire with 7 subscales. Frequency of slips and lapses (e.g., pertaining to measurements and documentation) and the quality of patient interactions (e.g., unfriendliness or impatience) due to work stress were assessed at follow-up with three items each (potential score ranges = 3–15). Potential intermediate factors at baseline included work engagement (i.e., vitality and dedication (UWES)), work satisfaction (COPSOQ), anxiety (GAD-2), depressiveness (PHQ-2), and self-reported health. We ran multivariable linear regression using z-standardized exposures to estimate unstandardized coefficients (B) and 95% confidence intervals (CI). Potential intermediate factors were added separately to the regression models. Attenuation of the association between exposure and outcome toward the null value (B = 0) was interpreted as mediation.

### Results

High reward and lack of resources were weakly associated with the frequency of slips and lapses (the Bs were -0.18 and 0.23, respectively; p<0.05), with little evidence of mediation. With the exception of low recognition, all unfavorable psychosocial working conditions predicted a higher frequency of poor interactions with patients (p-values<0.01). These associations were attenuated by work engagement, work satisfaction, and health outcomes.

**Data Availability Statement:** Data cannot be shared publicly because of to privacy concerns. Data are available from the Data Protection Office

of the Heinrich-Heine University (contact via datenschutz@hhu.de) for researchers who meet the criteria for access to confidential data.

**Funding:** This study was funded by Deutsche Forschungsgemeinschaft (grant number LO 1730/7-1) to Adrian Loerbroks (https://www.dfg.de/). The funders had no role in study design, data collection and analysis, decision to publish, or preparation of the manuscript.

**Competing interests:** I have read the journal's policy and the authors of this manuscript have the following competing interests: Adrian Loerbroks has presented findings related to the health and working conditions of medical assistants at meetings or workshops of professional associations or companies (i.e., ABF-Synergie GmbH) and has received honoraria. Viola Mambrey declares no conflict of interest. This does not alter our adherence to PLOS ONE policies on sharing data and materials.

**Abbreviations:** B, Unstandardized coefficients; CI, Confidence interval; COPSOQ, Copenhagen psychosocial questionnaire; ERI, Effort reward imbalance; GAD, Generalized anxiety questionnaire; MA, Medical assistant; PHQ, Patient health questionnaire; SD, Standard deviation; UWES, Utrecht work engagement scale; WHO, World Health Organization.

## Conclusion

We found mostly non-significant associations between adverse psychosocial working conditions and the frequency of slips and lapses. However, unfavorable psychosocial working conditions among MAs predicted a higher frequency of poor interaction with patients due to stress.

## Introduction

Health care workers report challenging working conditions including a high workload, poor collaboration within the team, low job control, poor leadership and low social support [1–4]. Poor working conditions have been found to be associated with a higher risk of poor wellbeing (i.e., physical and mental health e.g., depression, anxiety, burnout) [5, 6] and poorer quality of care (e.g., medical errors) [7–10]. Wellbeing can be understood based on either pathogenic conceptualizations (e.g., anxiety or depression) or salutogenic notions (e.g., work engagement) [11]. Wellbeing, in turn, is associated with the quality of care (e.g., medical errors, low patient satisfaction) [8, 12]. Wellbeing can therefore be assumed to act as an intermediate factor and to thereby (partially) explain associations between working conditions and quality of care [7]. These pathways have been proposed for the physician profession in a theoretical framework [2, 7]. That framework hypothesizes a direct and an indirect pathway: The direct pathway–i.e., the system approach—postulates that structural determinants such as adverse psychosocial working conditions become entrenched and thereby facilitate or hinder health care workers to perform effectively (e.g., to provide high-quality care) [13]. The indirect pathway assumes that (poor) wellbeing evolves as a consequence of psychosocial working conditions and in turn affects the quality of care as an intermediate factor [7].

Quality of care is distinguished by the World Health Organization (WHO) in terms of six dimensions: efficiency, access, equity, effectiveness, patient-centeredness (e.g., empathy, communication), and safety (i.e., minimizing health risks to which patients are exposed) [14]. Patient safety refers to an event that could have resulted or did actually result in harm to a patient (e.g., medical errors). Patient-centeredness revolves around patient involvement in their care process and a functioning social interaction between patient and health care workers [14]. Longitudinal studies among health care workers examining the relationship between psychosocial working conditions and quality of care are scarce, have mainly been carried out in hospital settings and have mainly focused on nurses and physicians [15–18]. Prior work has considered the relationship between adverse working conditions and patient safety, with patient safety being measured by medical errors [15], overall patient safety [16], a scale combining patient safety (i.e., errors) and general quality of work [17], and a recent study by our group measuring important medical errors [18]. The findings of those prospective studies were inconsistent with some observing direct relationships between working condition and quality of care indicators [17, 18] and some only indirect relationships [15, 16]. Notably, patient-centeredness has not been examined longitudinally as a distinctive dimension yet. Examination of potential intermediate factors of those relationships with pathogenic conceptualizations of wellbeing (i.e., poor mental health) yielded two studies finding a mediating effect [15, 18] and another study not [17]. Based on a salutogenic conceptualization of wellbeing (i.e., work engagement) one study suggested a potential mediating effect [18]. In order to provide an optimal quality of patient care in terms of its different dimensions, it is important to identify the specific working conditions, which may provide benefits to both patients in terms of improved quality of care and thus better outcomes, and to health care workers' wellbeing.

In Germany, outpatient care is a central pillar of the health care system and is provided by general physician practices, specialists practices and medical care centers [19]. The teams include general practitioners and/or specialists and medical assistants (MAs), with the latter representing the largest occupational group in outpatient care in Germany [20]. MAs have a broad range of responsibilities and tasks including administrative tasks (e.g., documentation of treatment, accounting), medical tasks (e.g., performing X-rays, CTs, injections, wound care, laboratory diagnostics and patient education) and the organization of the practice by scheduling appointments of patients and managing the reception desk of the practices [21]. MAs are usually the first contact for patients when they get in touch with the practice (e.g., via telephone or at the reception desk). Just like other occupational groups in the health care sector, MAs report unfavorable psychosocial working conditions (e.g., multitasking, low job control or poor collaboration) [9, 22].

MAs themselves were found to believe that their unfavorable psychosocial working conditions may result in minor errors (i.e., slips and lapses e.g., documentation or measurement errors) and poor social interactions with patients [23, 24]. Minor errors that do not result in harm for the patients' wellbeing are frequent in primary care [25]. However, if they go unnoticed they may lead to misdiagnosis and prescription errors, which in turn may result in harm to patients [25]. Further, a good interaction with patients in terms of communication is important as MAs are responsible for assessing the urgency of patients medical concerns (viz. telephone triage), for preparing the medical history of the patients and for providing patient education [21]. Effective interaction and communication between health care providers and patients has been found to improve health outcomes among patients (e.g., successful depression case management, safe telephone triage) [24, 26, 27], to increase patient satisfaction [28] and further to reduce safety lapses in primary care [29]. Previous work from our group among MAs found cross-sectional associations between adverse working conditions and a higher frequency of slips and lapses due to work stress [30]. Moreover, several unfavorable working conditions such as a high workload, low job control and poor collaboration showed pronounced associations with a higher frequency of poor interaction with patients due to work stress [30].

The set of important tasks of MAs described above highlights the contribution of MAs to the quality of patient care in terms of patient safety and patient-centeredness. To clarify the direction of association and potential causality prospective studies are needed. Previous longitudinal studies on the relationship between unfavorable working conditions and quality of care are sparse, mainly focus on professions working in inpatient health care [15–17] and have not specifically examined slips and lapses, which are the most common type of error, or patient-centeredness as a separate dimension of quality of care [15–18]. Accordingly, based on data from a professional group who mainly works in outpatient care (i.e. MAs) this study aimed i) to carry out an analysis of slips and lapses as well as for the first time ii) prospective analyses of the link between psychosocial working conditions and patient-centeredness. We thus drew on prospective data to examine the longitudinal relationships between adverse working conditions and slips and lapses as well as patient interaction among MAs. Further, we explored potential intermediate factors (i.e., salutogenic and pathogenic concepts of wellbeing) of these relationships among MAs in Germany.

## Materials and methods

### Study sample

In this study, we drew on data from a cohort study among MAs. At baseline (21.09.2016–05.04.2017) and follow-up (17.03.2021–27.05.2021) participants received a questionnaire as an online survey or on request as a hard-copy version. The nationwide recruitment was supported

by a number of associations and organizations, which distributed flyers, shared information on the study internally or on their respective homepage. The recruitment efforts are detailed elsewhere [9]. MAs were eligible when they were currently in training or held a MA degree. The baseline questionnaire was completed by 944 MAs. Invitations to participate at follow-up were sent to the participants via an e-mail or a post letter and reminders sent out after 3 and 6 weeks. In total, 537 MAs (56.9%) participated at follow-up. Detailed non-responder analyses were performed and presented elsewhere [18]. Briefly, follow-up participants were older at baseline, had more years of work experience, worked less often full time and displayed slightly higher depression baseline scores than non-participants. Differences regarding the psychosocial working conditions were not found. On average 4.40 years (standard deviation [SD] = 0.10) had elapsed between baseline and follow-up assessments [range: 4.04 years to 4.62 years]. Only those MAs who reported to be employed as MAs at both baseline and follow-up were included in the current longitudinal analysis (n = 408). Non-eligible MAs reported either current employment but not as a MA, parental leave, unemployment or retirement [18]. The study was approved by the ethics committee of the Medical Faculty of the Heinrich-Heine-University of Düsseldorf, Germany (baseline study: #4778; follow-up study: #2019–819). Written informed consent was obtained at baseline and follow-up from all individual participants included in the study. We drew on the STROBE (STrengthening the Reporting of Observational studies in Epidemiology, see: https://www.strobe-statement.org/) guidelines (see S1 Table) in writing this report.

## Questionnaire

**Determinants at baseline: Psychosocial working conditions.** Psychosocial working conditions were measured at baseline using the effort-reward imbalance (ERI) questionnaire [31] and a MA-specific working conditions questionnaire [9]. The ERI questionnaire contains 17 items, which can be grouped into two sub-dimensions, these are, effort [6 items, i.e., a high workload, time pressure and responsibility; potential score range = 6 to 24] and reward [11 items, i.e., high salary, high esteem and good career prospects; range = 11 to 44]. The items are presented as statements and responses are to be indicated on a 4-point Likert scale ranging from "strongly disagree" to "strongly agree". A higher score reflects a higher effort or higher reward, respectively. According to the ERI model, work-related distress is caused by an imbalance at an individual level between high effort spent and low reward received. The level of imbalance can be quantified using the ERI ratio. The ER-ratio is based on the division of the sum scores of effort and reward, multiplied by a correction factor which reflects the opposite number of items (correction factor in this study = 1.83). The ERI model proposes that an ERI ratio exceeding 1.0 indicates work stress.

MA-specific work stressors and resources were assessed by a questionnaire developed by our group. The items were created based on prior qualitative research [1], refined by cognitive interviews and the resulting questionnaire was psychometrically validated [9]. In total, 29 items measure 7 types of MA-specific work stressors or resources that are presented as statements and answered using a 4-point Likert scale (strongly disagree, rather disagree, rather agree, strongly agree) [9]. Each factor comprises 3–6 items that are used to calculate factor-specific sum scores with a higher score reflecting a higher exposure to the respective stressor. The 7 factors are: 1) workload, 2) job control, 3) collaboration with supervisor/colleagues, 4) gratification, 5) practice organization, 6) resources and 7) leadership behavior. A more detailed description of the factors can be found in S2 Table and elsewhere [9, 18].

**Potential intermediate factors at baseline: Wellbeing.** The following constructs were considered as indicators of wellbeing:

1. Work engagement captures a positive, fulfilling state of mind regarding one's work [32]. Work engagement was assessed by the 9-item Utrecht Work Engagement Scale (UWES), which consists of three dimensions: vigor [e.g., high work energy, work-related persistence; 3 items], dedication [e.g., inspiration, pride in work; 3 items] and absorption [e.g., feeling completely absorbed in one's work; 3 items] [32]. Previous studies indicated that vigor and dedication, in particular, determine the quality of health care rather than absorption [33, 34]. Therefore, the assessment of work engagement was limited to those two subscales in the current study. Responses are provided on a 7 point Likert scale ranging from "never" (0) up to "always" (6). Sum scores for vigor and dedication were calculated and divided by the number of items (3) [potential score range = 0–6 for each subscale].

2. A single item from the Copenhagen Psychosocial Questionnaire (COPSOQ) was used to measure work satisfaction [35] ("Regarding your work in general. How pleased are you with your job as a whole, everything taken into consideration?"). Responses are provided on a 4-point Likert scale (very unsatisfied, unsatisfied, satisfied, very satisfied) with a score range from 0–3.

3. Health was assessed in terms of mental health (i.e. anxiety and depressive symptoms) and general health. Anxiety and depressive symptoms were measured using the generalized anxiety disorder questionnaire (GAD-2) and patient health questionnaire (PHQ-2), respectively [36]. Items are presented as statements and provided on a 4 point Likert scale inquiring after the frequency of symptoms varying between "not at all" and "almost every day" (3). The instruments' scores range from 0 to 6, respectively. General health status (i.e., self-rated health) was measured by one item ("How is your health status in general?"). Responses were provided on a 5-point Likert scale varying from 1–5 (very good, good, average, bad, very bad) [9, 37].

**Outcomes at follow-up: Slips and lapses at work and poor interaction with patients.**
Quality of care was measured by a questionnaire that our group developed based on previous qualitative research [23]. That prior study revealed that MAs believe that stressful working conditions affect the quality of their work primarily in terms of slips and lapses as a well as poor interactions with patients. Items were refined based on cognitive interviews and psychometrically evaluated [30]. The questionnaire consists of 6 items that are presented as statements with responses provided on a Likert scale ranging from 1–5, reflecting the frequency (never, rarely, occasionally, most of the time, always). Factor analyses suggested 2 factors consisting of 3 items each [30]. Sum scores are calculated for each factor and divided by the respective number of items [potential score range = 3–15]. A higher score reflects a higher perceived frequency of the occurrence of the outcome (poor quality of care) due to work stress. The two factors are: (1) "slips and lapses", due to work stress during measurements on patients, related to patient records or documentation; (2) "poor interactions with patients", due to work stress in terms of unfriendliness, impatience, and perceived time pressure.

## Statistical analysis

There are several statistical approaches to estimate potentially causal effects in observational studies with continuous outcome variables. In a recent methodological paper, Tennant et al. [38] caution researchers to use change scores, as they may provide biased results. Two alternatives are presented, which rely on the utilization of the follow-up scores of the outcome variable either with or without adjustment for the outcome's baseline scores. The decision as to which approach to adopt is based on conceptual considerations, in particular the hypothesized

relationships between the key variables of interest, and the type of effect sought (total vs. direct). Within the study team, we intensively discussed which approach is the most suitable based on our research question and assumed relationships between variables. We agreed that the total effect is the relevant effect type given our research aim and decided to use only the follow-up scores of the outcome variables, that is, without adjustment for the outcomes' baseline scores.

For the primary statistical analysis, continuous baseline exposure variables (i.e., z-scores) (i.e., ERI variables, MA-specific working conditions) and continuous follow-up outcome variables (quality of care indicators) were used. Associations were quantified using unstandardized regression coefficients (B) and corresponding 95% confidence intervals (95% CIs), which were estimated based on multivariable linear regressions. Regression models were computed separately for each combination of exposure and outcome variable. Models initially remained unadjusted and were then adjusted for age and leadership position [34]. Sex as a potential confounding factor was excluded from the analysis due to a very low number of non-female participants ($n$ = 5, 1.23%). Due to conceptual overlap with the outcome "poor interaction with patients" the sub-dimension "resources" of the MA-specific questionnaire was removed from the analysis with the outcome "poor patient interaction" (the problematic items in the resources sub scale were: "I enjoy the interaction with patients" and "I enjoy the fact that my profession is a social activity") [30].

To explore potential intermediate factors, the corresponding variables were added separately to the regression models as continuous variables (i.e., vigor, dedication, depression, and anxiety) or ordinal variables (i.e., work satisfaction and self-rated health). Indication of mediation was considered to be present if the association between a given exposure and a given outcome was attenuated towards the null value (B = 0) after adjustment for the respective potential intermediate factor. This approach is referred to as the difference method [39].

Statistical analysis was performed using IBM SPSS Version 27.0. Missing values ranged from 0.0% (i.e., errors, vigor and work satisfaction) to 6.9% (i.e., ERI ratio) and were not imputed.

## Results

### Sample characteristics and descriptive results

Sample characteristics can be found in Table 1. The sample consisted almost entirely of female participants (98.8%) and had a mean age of 41.8 years (SD = 10.4). Half of the MAs reported to hold a leadership position (50%) and roughly 72% reported work stress according to the ERI ratio (i.e., ERI ratio >1.0). Scatterplots for the exposure variables and outcome variables can be found in S1 Fig.

As shown in Table 2, most of the examined adverse working conditions did not show significant associations with slips and lapses. Only high reward and a lack of social resources were significantly predictive of a higher frequency of slips and lapses due to work stress: a higher reward predicted a reduced frequency of slips and lapses [B = -0.18, 95%CI = 0.34, -0.02] and an increasing lack of resources was related to a higher frequency of slips and lapses [B = 0.23, 95%CI = 0.07, 0.40].

We observed significant relationships between all types of adverse psychosocial working conditions and a poor social interaction with patients, except for the MA-specific subscale gratification (see Table 3). For instance, an increasing ERI-ratio was related to a higher frequency of perceived poor social interaction with patients due to stress [B = 0.35, 95%CI = 0.14, 0.57]. Likewise, significant associations were observed for effort and reward [B = 0.31, 95% CI = 0.10, 0.51; and B = -0.32, 95%CI = -0.52, -0.11]. With regard to MA-specific work

**Table 1. Characteristics of the study population (n = 408\*).**

| Characteristic | | | |
|---|---|---|---|
| | $n$ | (%) | Missings in % |
| Sex | | | |
| Female (vs. male) | 401 | (98.8) | 0.5 |
| Leadership position | | | |
| Yes | 204 | (50.2) | 0.5 |
| Work stress according to ERI[a] (i.e., ratio >1.0) | | | |
| Yes | 273 | (71.8) | 6.9 |
| | M | (SD) | |
| Age | 41.8 | (10.4) | |
| Work experience (in years) | 19.4 | (11.0) | |
| Psychosocial working conditions | | | |
| Effort | 18.5 | (3.17) | 2.7 |
| Potential range 6–24 | | | |
| Reward | 28.7 | (6.04) | 4.2 |
| Potential range 11–44 | | | |
| ERI ratio[a] | 1.25 | (0.41) | 6.9 |
| (effort/reward)x1.83 | | | |
| MA[b] sub-scale (high) workload | 17.3 | (4.30) | 2.0 |
| Potential range 6–24 | | | |
| MA sub-scale (low) job control | 21.2 | (2.75) | 1.2 |
| Potential range 6–24 | | | |
| MA sub-scale (poor) collaboration | 8.20 | (2.82) | 1.5 |
| Potential range 4–16 | | | |
| MA sub-scale (low) gratification | 11.4 | (2.74) | 1.2 |
| Potential range 4–16 | | | |
| MA sub-scale (poor) practice organization | 6.52 | (2.06) | 0.2 |
| Potential range 3–12 | | | |
| MA sub-scale (lack of) resources | 4.55 | (1.64) | 1.7 |
| Potential range 3–12 | | | |
| MA sub-scale (poor) leadership behavior | 7.99 | (2.29) | 0.7 |
| Potential range 3–12 | | | |
| Slips and lapses | 5.45 | (1.58) | 0.2 |
| Potential range 3–15 | | | |
| Poor interaction with patients | 8.10 | (2.01) | 0.2 |
| Potential range 3–15 | | | |
| Mediators | | | |
| Vigor[c] | 3.52 | (1.34) | 0.0 |
| Potential range 0–6 | | | |
| Dedication[c] | 3.86 | (1.38) | 0.7 |
| Potential range 0–6 | | | |
| Depression[d] | 1.39 | (1.43) | 2.0 |
| Potential range 0–6 | | | |
| Anxiety[e] | 1.36 | (1.62) | 1.0 |
| Potential range 0–6 | | | |
| Characteristics | | $n$ (%) | Missing in % |

(*Continued*)

**Table 1.** (Continued)

| Work satisfaction[f] | | | | 0.0 |
|---|---|---|---|---|
| | Very unsatisfied | 10 | (2.45) | |
| | Unsatisfied | 89 | (21.8) | |
| | Satisfied | 262 | (64.2) | |
| | Very satisfied | 47 | (11.5) | |
| Self-rated health | | | | 1.5 |
| | Very good | 79 | (19.7) | |
| | Good | 174 | (43.3) | |
| | Average | 126 | (31.3) | |
| | Poor | 20 | (4.98) | |
| | Very poor | 3 | (0.75) | |

\*n with complete data on the respective variable and item; Mean (M); standard deviation (SD);

[a] effort-reward imbalance questionnaire (ERI); ERI = (Effort\*11)/(Reward\*6);

[b] medical assistant (MA);

[c] sub-dimension of the 9-item Utrecht Work Engagement Scale;

[d] Patient Health Questionnaire (PHQ-2);

[e] Generalized Anxiety Disorder questionnaire (GAD-2);

[f] Copenhagen Psychosocial Questionnaire.

**Table 2. Longitudinal associations of psychosocial working conditions (z-scores) at baseline with slips and lapses at follow-up among medical assistants (linear regression).**

| Characteristic | | Slips and lapses | | | |
|---|---|---|---|---|---|
| | | Model I[a] | | Model II[b] | |
| | | B[c] | 95%CI[d] | B | 95%CI |
| ERI model | | | | | |
| Effort | z-score[e] | -0.02 | (-0.18, 0.14) | -0.00‘ | (-0.16, 0.16) |
| Reward | z-score | -0.18 | (-0.34, -0.03)* | -0.18 | (-0.34, -0.02)* |
| ERI-ratio | z-score | 0.14 | (-0.03, 0.30) | 0.14 | (-0.02, 0.31) |
| MA-specific instrument | | | | | |
| Workload (high) | z-score | -0.04 | (-0.20, 0.11) | -0.03 | (-0.18, 0.12) |
| Job control (low) | z-score | -0.07 | (-0.22, 0.08) | -0.06 | (-0.21, 0.10) |
| Collaboration (poor) | z-score | 0.10 | (-0.05, 0.26) | 0.11 | (-0.05, 0.27) |
| Gratification (low) | z-score | 0.10 | (-0.05, 0.26) | 0.10 | (-0.06, 0.25) |
| Practice organization (poor) | z-score | 0.11 | (-0.05, 0.27) | 0.12 | (-0.04, 0.28) |
| Resources (lack of) | z-score | 0.22 | (0.06, 0.38)** | 0.23 | (0.07, 0.40)** |
| Leadership (poor) | z-score | 0.13 | (-0.03, 0.29) | 0.13 | (-0.03, 0.29) |

Effort-reward imbalance questionnaire (ERI) or medical assistant (MA)-specific work stress questionnaire; for each exposure variable a separate regression model was computed; indicators of the overall model fit of the regression models can be found in S3A Table

[a] unadjusted;

[b] adjusted for age and leadership position at baseline;

[c] unstandardized coefficient;

[d] confidence interval (CI);

[e] a higher score reflects a higher agreement to the respective stressor; ‘exact B = 0.004,

\*$p < 0.05$,

\*\*$p < 0.01$,

\*\*\*$p < 0.001$

**Table 3. Longitudinal associations of psychosocial working conditions (z-scores) at baseline with poor interaction with patients at follow-up among medical assistants (linear regression).**

| Characteristic | | Poor interaction with patients | | | |
|---|---|---|---|---|---|
| | | Model I[a] | | Model II[b] | |
| | | B[c] | 95%CI[d] | B | 95%CI |
| **ERI model** | | | | | |
| Effort | z-score[e] | 0.29 | (0.09, 0.50)** | 0.31 | (0.10, 0.51)** |
| Reward | z-score | -0.33 | (-0.53, -0.13)** | -0.32 | (-0.52, -0.11)** |
| ERI-ratio | z-score | 0.37 | (0.15, 0.58)** | 0.35 | (0.14, 0.57)** |
| MA-specific instrument | | | | | |
| Workload (high) | z-score | 0.32 | (0.13, 0.52)** | 0.33 | (0.14, 0.53)** |
| Job control (low) | z-score | 0.29 | (0.09, 0.49)** | 0.33 | (0.13, 0.53)** |
| Collaboration (poor) | z-score | 0.46 | (0.27, 0.66)*** | 0.44 | (0.24, 0.64)*** |
| Gratification (low) | z-score | 0.15 | (-0.05, 0.34) | 0.12 | (-0.08, 0.32) |
| Practice organization (poor) | z-score | 0.28 | (0.08, 0.48)** | 0.29 | (0.08, 0.49)** |
| Resources (lack of) | z-score | f | | f | |
| Leadership (poor) | z-score | 0.41 | (0.21, 0.61)*** | 0.41 | (0.21, 0.62)*** |

Effort-reward imbalance questionnaire (ERI) or medical assistant (MA)-specific work stress questionnaire; for each exposure variable a separate regression model was computed; indicators of the overall model fit of the regression models can be found in S3B Table;

[a] unadjusted;

[b] adjusted for age and leadership position at baseline;

[c] unstandardized coefficient;

[d] confidence interval (CI);

[e] a higher score reflects a higher agreement to the respective stressor;

[f] sub-scale "resources" removed from analysis from "poor interaction with patients" due to conceptual overlap;

*$p < 0.05$,

**$p < 0.01$,

***$p < 0.001$

stressors particularly pronounced relationships were observed for poorer collaboration [B = 044, 95%CI = 0.24, 0.64] and poorer leadership behavior [B = 0.41, 95%CI = 0.21, 0.62].

Mediation analysis was performed for both significant and non-significant associations between exposures and outcomes (see S4 Table). Focusing on significant associations from the primary analysis, we observed an attenuation towards the null value (B = 0) for the association of reward with slips and lapses after adjustment for work engagement and depression. No meaningful attenuation toward the null value was observed for the association of lack of resources with slips and lapses after adjustment for any of the intermediate factors (see Tables A and B in S4 Table). With regard to poor social interactions, the significant relationships with ERI dimensions (i.e., effort, reward, ERI-ratio) and the MA-specific work stressors with patients were overall attenuated towards the null value by all potential intermediate factors and particularly by work engagement and mental health (see Tables C and D in S4 Table).

## Discussion

We found mostly statistically non-significant associations between unfavorable psychosocial working conditions and slips and lapses. With the exception of higher reward and an increasing lack of resources, which were significantly associated with slips and lapses. The association of reward with slips and lapses may partially be explained by work engagement and depression. For the association with lack of resources no evidence of meaningful mediation was

observed. Except for low recognition, all adverse psychosocial working conditions considered were significantly related to a higher frequency of poor interactions with patients. An attenuation of these associations was observed due to each potential intermediate factors and in particular for work engagement and mental health.

## Comparison to prior research

Previous cross-sectional analyses by our group found strong positive associations of poor practice organization and a high workload with increased slips and lapses due to work stress [30]. These findings could not be reproduced in the current longitudinal analysis. We only found lower reward and a higher lack of social resources to be weakly, but significantly associated with an increasing frequency of slips and lapses. A direct comparison with previous longitudinal studies is challenging due to differences in the occupational groups studied, the health care settings or the differences in exposures (i.e., single job stress variable, focus on collaboration) or outcomes (i.e., no focus on minor errors) examined [15–18].

The relationship between adverse working conditions and patient safety indicators (i.e., medical errors, and overall patient safety) has been examined in four cohort studies [15–18]: one study was conducted among nurses in Japan and found a positive relationship between adverse working conditions (nursing stress scale, e.g. high workload, conflicts with supervisor or colleagues; combined into a single variable) and the self-reported frequency of near misses and adverse events (combined into a medical error risks variable) [15]. Another study found interpersonal teamwork, which corresponds to the collaboration sub-factor in our study, to be only indirectly predictive of clinician-rated overall patient safety among intensive care nurses and physicians in Switzerland [16]. Interpersonal teamwork was found to facilitate team organization and coordination behavior, which in turn was linked to positive clinician-rated patient safety [16]. A third study, carried out among physicians in Germany, found a link between self-reported psychosocial working conditions (i.e., social stressors and time pressure) and physicians' perception of impairment of their clinical work and quality of care, as measured with a scale combining patient safety (i.e., errors) and loss and/or reduction of quality. Increased social stressors (i.e., collaboration) and time pressure were related to a decrease in physician-perceived quality of care [17]. The fourth study is from our group and drew on the same cohort data as the current study thus focusing on the same occupational group (i.e., MAs) and the same psychosocial working conditions; however, the outcome studied differed and was dichotomous and addressed important medical errors rather than slips and lapses. Of all working conditions studied, poor collaboration was the only one significantly related to the concern to have made an important error [18].

In most of the abovementioned prospective studies, working conditions in general and in particular collaboration were found to be predictive of poor patient safety (e.g., medical errors and overall patient safety) [15–17], which is not in line with our results. There is no straightforward explanation as to why specifically reward and lack of social resources should be linked with slips and lapses in MAs rather than other and apparently more plausible types of work stressors that affect organization and autonomy (e.g., practice organization and job control). We cannot rule out that our findings are random. Slips and lapses are defined as execution failures based on failure in attention and memory [40]. One might assume that memory and attention, and thereby slips and lapses, are immediately affected by the work stressors rather than occurring with a great time lag or as a result of work stress accumulation over time. If that held true, removing the stressor would directly improve the outcome (i.e., lower frequency of slips and lapses) [41]. However, in that case, the time lag between baseline and follow-up may have been too long and the change in working conditions within the participants too

great to detect the potential direct and immediate effect of adverse working conditions on the frequency of slips and lapses, thereby underestimating the true causal effect [41, 42]. This is in line with the maximum time lag of 12 months in the prior prospective studies, with the exception of our group's previous study [18]. Those earlier studies found significant associations between baseline assessment of working conditions and follow-up assessment of patient safety [15–17], compared to the time lag of 4 years in the present study.

We found a pattern of consistent significant relationships between a wide range of working conditions (e.g., poor collaboration, high workload) and social interaction with patients. A particularly strong relationship was found for poorer collaboration and for poorer leadership behavior and a reported increasing frequency of poor patient interaction due to work stress. To the best of our knowledge, no cohort study examined psychosocial working conditions and patient-centeredness in particular (i.e., measuring the latter by social interaction with patients or patient satisfaction).

Collaboration among MAs and with their supervisors depends to a large extent on communication and coordination (e.g., division and delegation of tasks) [1, 24, 43]. A qualitative study among nurses in the Netherlands focused on nurses' work environment and how it affects patient experiences of quality of care. Participants reported that teamwork relies on communication within the team and is necessary to provide clarity and composure towards the patients [44]. Moreover, a cross-sectional study among health care workers in intensive care units in the United Kingdom found that increasing perceived openness in communication (e.g., speaking openly, accuracy of information shared) among team members was associated with the extent to which individuals reported to understand the care needs of their patients [45]. These findings highlight the importance of good collaboration within the team and subsequent communication between patients and MAs to ensure appropriate patient care.

Furthermore, due to the practice structure in outpatient care settings in Germany, which is characterized by mostly small teams, MAs are particularly dependent in the execution of their tasks and interaction with patients on their supervisor in terms of leadership style and personality [1]. Some cross-sectional studies show that certain styles of leadership are associated with increased patient satisfaction [46]. One study suggests that positive leadership, based on leader-follower interactions, provides health care workers with orientation and clarification of tasks as well as procedures, which in turn facilitates patient care [47]. Our results add that poor leadership behavior predicts an increase in the frequency of poor interaction with patients among MAs. Overall, our findings highlight the link between adverse working conditions (in particular collaboration and behavior of supervisor) among MAs and subsequent poor interaction with patients.

The model proposed by Angerer and Weigl (2015) hypothesizes that working conditions can affect wellbeing [i.e., salutogenic concepts (e.g., work engagement, work satisfaction) or pathogenic concepts (e.g., poor mental and overall health)] and in turn exert effects on the quality of care [7]. In this study we found that work engagement may partially explain the link between reward and slips and lapses. Further, we found work engagement, work satisfaction and health (i.e., mental and overall health) to partially explain the link between working conditions and the interaction with patients. Working conditions among MAs may result on the long term in job satisfaction [3] and work engagement, which is characterized by a general positive attitude towards work and consequently higher productivity, more energy, and enthusiasm [7, 32]. These feelings may increase concentration and thereby minimize slips and lapses, and enhance social interactions with patients by directing more attention to the patient and their thoughts, wishes and feelings [48, 49]. Moreover, high reward as favorable working conditions, may lower the risk of depression [5], which in turn may decrease frequency of slips and lapses among MAs, through higher concentration.

However, unfavorable working conditions are also associated with pathogenic notions of wellbeing among MAs in terms of poor mental (i.e., anxiety and depression) and physical health (i.e., overall health) [9]. Further, poor wellbeing may manifest itself in the form of emotional detachment, fatigue and lower empathy [7, 50], which in turn may present obstacles to interaction with patients among MAs. Our findings are supported by a prospective study in Japan, which found higher depression scores to be an intermediate factor between job stressors and perceived risk of medical errors due to a reduced attention among nurses [15]. By contrast, a prospective study among hospital physicians did not find pathogenic concepts of wellbeing (i.e., work-related mental strain) to mediate the relationship between job demands (i.e., social stressors, patient demands and time pressure) and physician-rated overall quality of care [17]. However, the authors assume that the one year time lag between their assessments was too short to observe a true effect of the working conditions on the strain of the participants [17]. Our exploratory results suggest a possible intermediate position of salutogenic and pathogenic wellbeing concepts between the relationship of adverse working conditions among MAs and poor interaction with patients. In summary, we add prospective data to the scarce literature and to the best of our knowledge, these are the first prospective analyses focusing on the relationship between working conditions and patient-centeredness (i.e., interaction with patients) as a dimension of quality of care among MAs.

## Methodological considerations

The strengths of our study are its prospective design, the assessment of a broad range of psychosocial working conditions and of different constructs of quality of care (i.e., patient safety and patient-centeredness).

Due to the widespread recruitment effort for the baseline assessment the response rate cannot be calculated, and selection bias cannot be ruled out. However, the MAs who participated in this study are fairly representative of the general MA population in Germany with regard to gender, age and employment status [20]. Moreover, we performed an non-responder analyses for MAs who have vs. who have not participated at follow-up, with no significant differences with regard to the predictor variables (e.g., psychosocial working conditions,) [18], and the measure of patient-centeredness and only a marginal difference with regard to the mean values of slips and lapses between follow-up participants and non-participants (mean values: 5.6 vs. 5.3; p<0.01).

We did not collect data on the MAs' employer and thus do not know whether we included MAs who work for the same employer. In case of such clustering of observations, different statistical approaches had been preferable (e.g., multilevel modeling). However, such clustering to a large extent seems unlikely as we recruited our sample of 408 MAs from all over Germany and not primarily through their employer. The distribution of some variables may have displayed a restricted range (e.g., slips and lapses, work satisfaction), which may have impacted the regression estimates. Additionally, we measured the frequency of quality of care indicators due to work stress by self-reports. We cannot rule out that participants may have disclosed only low frequencies of slips and lapses due to social desirability (no open error culture), due to a lack of reflection due to a high workload (i.e., lack of time) or in fear of negative consequences [51]. Furthermore, slips and lapses often do not translate into serious adverse events and an increase in frequency due to work stress may have remained partially unnoticed by participants. In addition, slips and lapses which were directly corrected by the MA or the supervisor may not have been recalled as such. Moreover, it is possible that the outcome "interaction with patients" is more strongly associated with emotional responses than slips and lapses and

is therefore easier to perceive and report. In addition, the quality of care indicators were assessed as "the areas of patient care that are impacted by work stress", thereby providing a reference to the exposures. This may have led to an underestimation of slips and lapses and patient interaction as work stress among MAs is known to be high [9] and participants may be used to a certain work stress level and only reported the frequency of quality of care indicators based on extreme work stress situations.

In this study, prospective data were assessed at two time points. A third assessment would have been desirable to disentangle wellbeing as intermediate factors of the relationship between adverse working conditions and poor interaction [52]. In addition, a shorter time lag between the assessments may have been better to capture the relationship between adverse working conditions and slips and lapses. However, a shorter time lag may have been less suitable to observe adverse working conditions' effects on wellbeing and further on poor interaction with patients [41].

Finally, the differentiation between mediators and cofounders is based on conceptual considerations [53]. Therefore, the attenuation of observed associations could also be due to confounding rather than mediation.

## Recommendations for practice, prevention, and future research

Our findings need to be supported by further prospective studies. Those studies should extend self-reported measurements of slips and lapses to more objective assessments as already performed in hospital settings [54, 55], for example with observer-based ratings to validate the self-reported quality of care indicator. To examine potential direct and causal effects of adverse working conditions with slips and lapses a shorter time lag between the assessments should be applied. In addition, a study with more than two waves is needed to better understand potential mediating effects of wellbeing between the link of unfavorable working conditions and poor interaction with patients. Moreover, in an exploratory approach, we applied the difference method to explore potential intermediate factors. Future research could examine potential mediation in greater detail [56].

Almost all examined work stressors were predictive of an increasing frequency of poor interactions with patients. This highlights that improvements are needed at the personal, practice and systematic levels. First of all, at the political level, it should be acknowledged that the working conditions among MAs are poor [9] and that the shortage of skilled MAs increasingly worsens the working conditions of the remaining MAs [57, 58]. Political structures should be adjusted to increase the attractiveness of the MA profession and improve retention. For instance, through legislation to pay MAs according to collective agreement or incentives for employing physicians to pay according to collective agreement. Moreover, we suggest that a promising starting point is to improve communication across hierarchical structures. Courses on leadership skills could be included in the medical school curriculum or made mandatory in continuing education for practicing physicians. On a practice level, team collaboration can be strengthened through regular team meetings, an open communication culture, and team activities [59, 60]. On a personal level, courses in active listening could improve communication skills of MAs with patients. This could help MAs to better focus on the patients' need during times of work stress [61]. Although findings for the relationship between work stressors and slips and lapses were largely non-significant, a positive feedback culture based on openness and transparency within the work team, rather than a blame culture, may promote awareness of slips and lapses and their subsequent reporting [51].

## Conclusion

We present prospective data for an association which remained largely unexamined. In summary, we found mostly non-significant associations between adverse working conditions and slips and lapses. However, unfavorable psychosocial working conditions among MAs significantly predicted a higher frequency of poor interaction with patients. These significant relationships were partially mediated by work engagement, job satisfaction, and health.

## Supporting information

**S1 Fig. Scatterplots.**
(PDF)

**S1 Table. STROBE checklist.**
(PDF)

**S2 Table. Detailed item description.**
(PDF)

**S3 Table. Overall model fit primary analysis.**
(PDF)

**S4 Table. Potential intermediate factors.**
(PDF)

## Acknowledgments

We want to thank all medical assistants who have participated in our study. Furthermore, we would like to thank Dr. Annegret Dreher (Institute of Occupational, Social and Environmental Medicine, Centre for Health and Society, Medical Faculty and University Hospital Düsseldorf, Heinrich Heine University Düsseldorf, Düsseldorf, Germany) for her valuable contributions to the discussion of the statistical approach.

## Author Contributions

**Conceptualization:** Adrian Loerbroks.

**Data curation:** Viola Mambrey.

**Formal analysis:** Viola Mambrey.

**Funding acquisition:** Adrian Loerbroks.

**Investigation:** Viola Mambrey.

**Methodology:** Viola Mambrey, Adrian Loerbroks.

**Project administration:** Adrian Loerbroks.

**Resources:** Adrian Loerbroks.

**Supervision:** Adrian Loerbroks.

**Writing – original draft:** Viola Mambrey.

**Writing – review & editing:** Adrian Loerbroks.

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
