## [Decision Letter · Decision Letter 0]

8 Nov 2023

PONE-D-23-29754Psychosocial working conditions as determinants of slips and lapses, and poor social interactions with patients among medical assistants in Germany: a cohort studyPLOS ONE Dear Dr. Mambrey, Thank you for submitting your manuscript to PLOS ONE. After careful consideration, we feel that it has merit but does not fully meet PLOS ONE’s publication criteria as it currently stands. Therefore, we invite you to submit a revised version of the manuscript that addresses the points raised during the review process.

If applicable, we recommend that you deposit your laboratory protocols in protocols. io to enhance the reproducibility of your results. Protocols.io assigns your protocol its own identifier (DOI) so that it can be cited independently in the future. For instructions see: https://journals.plos.org/plosone/s/submission-guidelines#loc-laboratory-protocols. Additionally, PLOS ONE offers an option for publishing peer-reviewed Lab Protocol articles, which describe protocols hosted on protocols.io. Read more information on sharing protocols at https://plos.org/protocols?utm_medium=editorial-email&utm_source=authorletters&utm_campaign=protocols.

We look forward to receiving your revised manuscript.

Kind regards,

Collins Atta Poku

Academic Editor

PLOS ONE

Journal Requirements:

"I have read the journal's policy and the authors of this manuscript have the following competing interests:Adrian Loerbroks has presented findings related to the health and working conditions of medical assistants at meetings or workshops of professional associations or companies (i.e., ABF-Synergie GmbH) and has received honoraria. Viola Mambrey declares no conflict of interest."

Reviewers' comments:

Reviewer's Responses to Questions

**Comments to the Author**

1. Is the manuscript technically sound, and do the data support the conclusions?

Reviewer #1: Yes

Reviewer #2: Yes

Reviewer #3: Partly

2. Has the statistical analysis been performed appropriately and rigorously? 

Reviewer #1: I Don't Know

Reviewer #2: Yes

Reviewer #3: No

3. Have the authors made all data underlying the findings in their manuscript fully available?

Reviewer #1: Yes

Reviewer #2: Yes

Reviewer #3: No

4. Is the manuscript presented in an intelligible fashion and written in standard English?

Reviewer #1: Yes

Reviewer #2: Yes

Reviewer #3: Yes

5. Review Comments to the Author

Reviewer #1: The authors have investigated a subject of great importance to the quality of interaction between clinicians and patients. The study has been well presented. You may want to read lines 509 and 510 to clarify the intended point. Congratulations

Reviewer #2: None

Reviewer #3: This is a very interesting cross-sectional research on potential associations for working conditions and slips/lapses for medical assistants. This was an extremely difficult study to execute, even moreso with the pandemic. The paper is very well-written and nicely mentions the need for more objective measures of slips and lapses. My main comments are about the analyses and the the possibility there could be restriction of range in the data, particularily for slips and lapses.

Major Comments

1) Regression models

a) Overall model fit: Suggest using the F-ratio, p-value, and adjusted R2

b) Correlation matrix: I recommend showing all the correlations among variables before the regression models. My concern is there could be high collinearity in the regression models.

c) Mediation analysis: "Mediation analysis was performed for significant associations between exposure and outcomes." This may be problematic because mediation can still occur with non-significant associations, see https://www.ncbi.nlm.nih.gov/pmc/articles/PMC2819361

- I'm definitely not an expert in mediation, but my limited understanding is the current best practice is to use bootstrapping: https://www.processmacro.org/index.html

Preacher and Hayes have many publications about this- https://www.processmacro.org/papers.html

d) Unit of analysis: It appears that each medical assistant was treated as an independent unit of analysis? Were there medical assistants from the same clinics, same regions? If so, cluster adjusted standard errors/variance should be used (or multilevel modeling). This may or may not make much of a difference in the results. I'm not sure about using cluster adjustment with mediation.

2) Restriction of range? Could limited ranges of responses be impacting the regression results? For example, for work satistifaction: only 2.45% of MA were very unsatisified and only 11.5% were very unsatisified. Slips and lapses appeared to be at the lower end of the range [mean = 5.45 with a min= 3], so they are at least somewhat rare? If this is the case, acknowledging it as a limitation should be sufficient.

Minor Comments

1) Graphs: It'd be really helpful to see scatter plots among the different variables. Perhaps this could be reported in the appendix?

2) Data availability: I understand privacy concerns, but would partial data-sharing be possible? Such as the individual survey data without demographic and/or other information about the Medical Assistants that might be used for re-identification?

6. PLOS authors have the option to publish the peer review history of their article (what does this mean?). If published, this will include your full peer review and any attached files.

Reviewer #1: No

Reviewer #2: No

Reviewer #3: No

---

## [Author Response · Author response to Decision Letter 0]

5 Dec 2023

Please see uploaded document "responses to reviewers".

---

## [Decision Letter · Decision Letter 1]

21 Dec 2023

Psychosocial working conditions as determinants of slips and lapses, and poor social interactions with patients among medical assistants in Germany: a cohort study

PONE-D-23-29754R1

Dear Dr. Mambrey,

We’re pleased to inform you that your manuscript has been judged scientifically suitable for publication and will be formally accepted for publication once it meets all outstanding technical requirements.

Kind regards,

Collins Atta Poku

Academic Editor

PLOS ONE

Additional Editor Comments (optional):

Reviewers' comments:

Reviewer's Responses to Questions

**Comments to the Author**

1. If the authors have adequately addressed your comments raised in a previous round of review and you feel that this manuscript is now acceptable for publication, you may indicate that here to bypass the “Comments to the Author” section, enter your conflict of interest statement in the “Confidential to Editor” section, and submit your "Accept" recommendation.

Reviewer #2: All comments have been addressed

Reviewer #3: All comments have been addressed

2. Is the manuscript technically sound, and do the data support the conclusions?

Reviewer #2: Yes

Reviewer #3: Yes

3. Has the statistical analysis been performed appropriately and rigorously? 

Reviewer #2: Yes

Reviewer #3: Yes

4. Have the authors made all data underlying the findings in their manuscript fully available?

Reviewer #2: Yes

Reviewer #3: No

5. Is the manuscript presented in an intelligible fashion and written in standard English?

Reviewer #2: Yes

Reviewer #3: Yes

6. Review Comments to the Author

Reviewer #2: None

Reviewer #3: The authors thoroughly addressed my comments. They also clearly responded to why some of my comments were not or could not be addressed.

This is an excellent paper. I recommend acceptance of the revised manuscript.

7. PLOS authors have the option to publish the peer review history of their article (what does this mean?). If published, this will include your full peer review and any attached files.

Reviewer #2: No

Reviewer #3: No

---

## [Editor Report · Acceptance letter]

8 Jan 2024

PONE-D-23-29754R1 

PLOS ONE

Dear Dr. Mambrey, 

I'm pleased to inform you that your manuscript has been deemed suitable for publication in PLOS ONE. Congratulations! Your manuscript is now being handed over to our production team.

Kind regards, 

on behalf of

Dr. Collins Atta Poku 

Academic Editor

PLOS ONE